# Box–Behnken Design (BBD)-Based Optimization of Microwave-Assisted Extraction of Parthenolide from the Stems of *Tarconanthus camphoratus* and Cytotoxic Analysis

**DOI:** 10.3390/molecules26071876

**Published:** 2021-03-26

**Authors:** Perwez Alam, Nasir Ali Siddiqui, Md. Tabish Rehman, Afzal Hussain, Ali Akhtar, Showkat R. Mir, Mohamed Fahad Alajmi

**Affiliations:** 1Department of Pharmacognosy, College of Pharmacy, King Saud University, Riyadh 11451, Saudi Arabia; nsiddiqui@ksu.edu.sa (N.A.S.); mrehman@ksu.edu.sa (M.T.R.); afihussain@ksu.edu.sa (A.H.); 2IT & Quality Unit, College of Pharmacy, King Saud University, Riyadh 11451, Saudi Arabia; aakhtar@ksu.edu.sa; 3Phyto-Pharmaceutical Research Laboratory, School of Pharmaceutical Education and Research, Jamia Hamdard, New Delhi 110062, India; showkatrmir@gmail.com

**Keywords:** Box–Behnken design, parthenolide, *T. camphoratus*, HPTLC analysis, cytotoxicity

## Abstract

Parthenolide, a strong cytotoxic compound found in different parts of *Tarchonanthus camphoratus* which motivated the authors to develop an optimized microwave-assisted extraction (MEA) method using Box–Behnken design (BBD) for efficient extraction of parthenolide from the stem of *T. camphoratus* and its validation by high-performance thin-layer chromatography (HPTLC) and cytotoxic analysis. The optimized parameters for microwave extraction were determined as: 51.5 °C extraction temperature, 50.8 min extraction time, and 211 W microwave power. A quadratic polynomial model was found the most suitable model with *R*^2^ of 0.9989 and coefficient of variation (CV) of 0.2898%. The high values of adjusted *R*^2^ (0.9974), predicted *R*^2^ (0.9945), and signal-to-noise ratio (74.23) indicated a good correlation and adequate signal, respectively. HPTLC analyzed the parthenolide (R_f_ = 0.16) content in *T. camphoratus* methanol extract (TCME) at λ_max_ = 575 nm and found it as 0.9273% ± 0.0487% *w*/*w*, which was a higher than expected yield (0.9157% *w*/*w*). The TCME exhibited good cytotoxicity against HepG2 and MCF-7 cell lines (IC_50_ = 30.87 and 35.41 µg/mL, respectively), which further supported our findings of high parthenolide content in TCME. This optimized MAE method can be further applied to efficiently extract parthenolide from marketed herbal supplements containing different *Tarconanthus* species.

## 1. Introduction

Parthenolide (sesquiterpene lactone, Figure 1) is a potent cytotoxic agent [1] and is present in large amounts in *Chrysanthemum parthenium* (feverfew; 85% *w*/*w*). Parthenolide reportedly prevents platelet clumping and serotonin and inflammatory mediators release [2]. Also, parthenolide has been demonstrated to possess anticancer and cytotoxic activities against several human cancer cell lines, including HeLa cells (cervical carcinoma), h9c2 cells (cardiomyoblasts), HT-29 cells (colon cancer), PC-3 and DU−145 cells (prostate cancer), and MDA-MB-231 and MCF7 cells (breast cancer) [3,4,5,6]. Therefore, the authors of this study sought to explore alternatives to feverfew for parthenolide content. *Tarchonanthus camphoratus*, belonging to the Asteraceae family, is considered to be a prominent substitute of feverfew for parthenolide content, especially in its stem and leaves [7].

Uneven distribution of phytoconstituents in different plant parts, degree of solubility of these phytoconstituents in various solvents, and extraction temperature require optimization to extract the desired phytoconstituents better. Sesquiterpene lactones such as parthenolide were first extracted in 1970 using the conventional extraction method of chloroform and petroleum-ether extraction solvents [8]. Subsequently, several other extraction methods were developed for parthenolide extraction, such as high-performance liquid chromatography (HPLC) [9] and the Soxhlet extraction method [10]. Recently, a progressive change has been observed in extraction technology to develop simple and effective sample preparation methods. The microwave-assisted extraction (MAE) technique, being inexpensive, simple, and efficient, is a promising technique for increasing the extraction of compounds from plants [11]. A substantial increase has been achieved in the yields of medicinal plant extraction by using microwave irradiation. In the event of microwave irradiation on natural material, electromagnetic waves are absorbed selectively by media with a high dielectric constant, resulting in more effective heating. In the course of absorption, the microwaves’ energy is converted into kinetic energy, thus allowing the selective heating of the microwave-absorbent plant material. The subsequent volume increase causes cells to explode, releasing their content into the liquid phase. When the liquid absorbs the microwaves, its molecules’ kinetic energy increases; consequently, the diffusion rate increases, resulting in faster mass transfer [12].

Response surface methodology (RSM) is an efficient method for optimizing the complex extraction processes comprising numerous variables [13]. The Box–Behnken design (BBD) of RSM operates at three levels (low, medium and high) and requires few experiments [14]. Several research groups have reported the various methods used to analyze parthenolide in Feverfew such as infrared spectrometry [15] and HPLC [16,17].

In this study, the authors applied the BBD method to optimize parthenolide extraction from *T. camphoratus* stems using MAE technique with three extraction variables (extraction temperature, extraction time and microwave power). The obtained BBD optimized extract was analyzed for parthenolide content by high-performance thin-layer chromatography (HPTLC) and assayed against two cancer cell lines (HepG2 and MCF-7).

## 2. Materials and Methods

### 2.1. Plant Material

The *T. camphoratus* stems (voucher specimen #15451) were collected from Wadi gamma (Saudi Arabia) and authenticated by field taxonomist Dr. Mohamed Yusuf. Post collection, the plant part was cut into small pieces, washed and dried in the plant drying room, and stored in a clean glass jar. A specimen was deposited in herbarium of Pharmacognosy Department, Pharmacy College, KSU, Saudi Arabia.

### 2.2. Apparatus and Reagents

The standard compound parthenolide (≥98%) was procured from Sigma-Aldrich (St. Louis, MI, USA). Methanol, ethyl acetate, and *n*-hexane of analytical grade were procured from WINLAB (Unit 13, Courtyard Workshops, Bath Street, Market Harborough LE16 9EJ, UK) and DMSO used in the cytotoxic analysis was procured from Sigma-Aldrich (St. Louis, MI, USA). Silica gel 60F254-precoated HPTLC plates (20 × 10 cm) were purchased from Merck (Darmstadt, Germany), and standard compounds along with extracts were applied (band wise) to plates using a CAMAG automatic TLC sampler-4 (ATS-4) and developed in a CAMAG automatic development chamber (ADC 2). Finally, the developed plates were documented using a CAMAG TLC Reprostar 3 and scanned using a CAMAG ATS 4 (CATS 4; CAMAG, Muttenz, Switzerland).

### 2.3. Microwave-Assisted Extraction of T. camphoratus Stems

The extraction of *T. camphoratus* (TC) stem powder (1 g) was performed in a closed container in a microwave (MARS 5, Matthews, NC, USA) using methanol as solvent. After the completion of the extraction procedure, the *T. camphoratus* methanol extract (TCME) was cooled and filtered. The residue was washed again with methanol thrice to get the final volume of TCME, filtered through a syringe filter (PTFE membrane, 0.45 µm, Phenomenex) and dried at reduced pressure using rotavapor (R-300, Buchi, Switzerland) to achieve the final percentage yield.

### 2.4. High-Performance Thin-Layer Chromatography (HPTLC) Analysis of TCME

#### 2.4.1. TLC Instrumentation and Chromatographic Conditions

In the process of analysis of parthenolide content in TCME by HPTLC methods, the TCME samples were applied on a 20 × 10 cm glass-backed HPTLC plate (coated with silica gel 60 F_254_) with a Hamilton Gastight Syringe (25 µL) fitted in Automatic TLC Sampler-4 at a speed of 160 nL/s. Afterward, the plate development took place in previously saturated ADC 2 for 20 min at 22 °C, with a mobile phase of *n*-hexane and ethyl acetate at the ratio of 3:1 (*v*/*v*). Further, the developed plate’s derivatization was done using reagent *p*-anisaldehyde, then the plate was dried gently, and scanned with CATS 4 (slit dimension: 4.00 × 0.45 mm; speed: 20 mm/s).

#### 2.4.2. Preparation of the Standard Stock Solution

A stock solution of parthenolide (1 mg/mL) was prepared by dissolving 1 mg of parthenolide in 1 mL of methanol and further diluted with methanol to make a final working concentration of 0.1 mg/mL. For calibration, 1–7 µL of working standard solution was applied to the HPTLC plate to provide a 100–700 ng/band concentration range.

#### 2.4.3. HPTLC Method Development and Validation

Several combinations of different solvents were used to select the most appropriate mobile phase to obtain the best resolution on chromatograms of standard and test extracts. The same mobile phase [*n*-hexane:ethyl acetate (3:1, *v*/*v*)] was used for efficient separation of various constituents of TCME samples. The developed HPTLC method was validated according to the International Council for Harmonisation of Technical Requirements for precision, accuracy, LOD (limit of detection), LOQ (limit of quantification), and robustness [18]. The LOD and LOQ for parthenolide were calculated by using the following equations 1 and 2, respectively:LOD = (3.3 × SD)/α(1)
LOQ = (10 × SD)/α(2)
where SD is the least standard deviation and α is the slope of the curve.

#### 2.4.4. Quantitative Analysis of Parthenolide in TCME Samples

For quantitative analysis, the TCME samples along with standards were spotted on HPTLC plates, and the parthenolide content (% *w*/*w*) in TCME samples was determined by measuring the area.

### 2.5. BBD Experimental Design

#### 2.5.1. Single-Factor Experimental Design

The range of different extraction variables (extraction temperature, time and microwave power) was selected based on observation of single-factor effect on the total extraction yield, which was used to optimize all extraction variables by Box–Behnken design (BBD) method to obtain the maximum parthenolide content from TCME. The impact of a single factor on total extraction yield was evaluated by varying the extraction temperature (30–90 °C), extraction time (20–80 min), and microwave power (50–600 W) while keeping extraction temperature (40 °C), extraction time (40 min), and microwave power (300 W) constant when another factor is variable.

#### 2.5.2. Optimization of Extraction Variables Using BBD Method

A 3 factorial (3^3^) Box–Behnken design (BBD; Design Expert Software, Trial version 12, Stat-Ease Inc., Minneapolis, MN, USA) of Response Surface Methodology (RSM) was used to optimize the three independent variables, namely, extraction temperature (*P*_1_), extraction time (*P*_2_), and microwave power (*P*_3_), with low (−1), medium (0), and high levels (+1) for each variable (Table 1). The BBD method had generated 17 experimental runs comprising five central points (Table 2). The response value was denoted by the total yield of parthenolide (*R*), and the results were fit into a second-order polynomial Equation (3):(3)$$\gamma =q_{0}+\overset{n}{\underset{i=1}{\sum }}q_{i}P_{i}+\overset{n-1}{\underset{\begin{array}{c} i=1\\ j\;>\;1 \end{array}}{\sum }}\overset{n}{\underset{j=2}{\sum }}q_{ij}P_{i}P_{j\;}+\overset{n}{\underset{i=1}{\sum }}q_{ii}P_{i}^{2}\;\;\;\;$$
where *Y* = Total parthenolide yield (*R*), *q*_0_, *q_i_*, *q_ii_*, and *q_ij_* = the regression coefficients of the intercept, linearity, square, and interaction, respectively. *P_i_*, *P_j_* = independent variables.

To interpret the effects of all independent variables on parthenolide yield, three-dimensional response surface plots were constructed. The “biggest-is-best” principle was applied for each response to maximize the outcome of parthenolide extraction. Finally, to confirm the study, three parallel experiments were carried out using the optimal extraction conditions with the maximum interest for practically authenticating the quality characteristic’s progress.

#### 2.5.3. BBD Model and Validity Testing

Box–Behnken design of response surface methodology was used for the analysis of the experimental results and optimization of three independent extraction variables, extraction temperature (*P*_1_), extraction time (*P*_2_), and microwave power (*P*_3_); *p*-values ≤ 0.05 were considered to be significant. The extraction of a sample using optimized independent extraction variables was carried out in triplicate (*n* = 3), and the experimental yield was compared with predicted values for model validation.

### 2.6. Cell Culture and Cytotoxicity Assay of BBD Run TCME Samples

In this study, two different human cancer cells MCF-7 (breast) and HepG2 (liver), were used. The cells were maintained in Dulbecco’s Modified Eagle Medium (DMEM) supplemented with bovine calf serum (10%; Invitrogen, Carlsbad, CA, USA) and 1% penicillin-streptomycin (Invitrogen, Carlsbad, CA, USA) at 37 °C with 5% CO_2_ supply in a humid chamber. The MCF-7 and HepG2 cells were plated (1 × 10^5^ cells/mL/well) in 24-well tissue culture plates 1 day before treatment. Stocks of TCME (*n* = 17) were prepared first in 100 μL DMSO (Sigma, St. Louis, MI, USA), and then in DMEM (100 mg/mL) following reconstitution of four working concentrations (100, 50, 25, and 10 μg/mL) in DMEM. The final concentration of DMSO never exceeded 0.1% in the treatment doses. After 24 h, the cells were treated with triplicated doses of each extract, including an untreated control (0.1% DMSO), and incubated for 48 h at 37 °C. Next, 100 µL of MTT (5 mg/mL; TACS MTT Cell Proliferation and Viability Assay Kit, BioTekhne, Minneapolis, MN, USA) was added to each well and incubated for 24 h. After incubation, 1 mL of 0.01 N HCL/isopropanol was added to the wells to solubilize the formazan on the shaker for 10 min. The absorbance of converted MTT was measured at λ = 490 nm with a microplate reader (Bio-Tek, Winooski, VT, USA). Vinblastine was used as a positive control. For each extract tested, the IC_50_ (concentration of tested compound needed to inhibit cell growth by 50%) was generated from the dose-response curves.

### 2.7. Statistical Analysis

All experiments were carried out with three independent replicates and values are presented as mean ± standard error of the mean (SEM). Data were statistically analyzed using the Student’s *t*-test to compare the means applying a significance level of *p* < 0.05.

## 3. Results

### 3.1. Effect of Single-Factor Tests on the Total Extraction Yield of T. camphoratus Stems

#### 3.1.1. Extraction Temperature Effect

To investigate extraction temperature effect on total extraction yield, the microwave temperature was varied from 30 °C to 90 °C, while the extraction time (40 min) and microwave power (300 W) were kept constant. It is clear from Figure 2A that the total extraction yield obtained at 30 °C was the lowest and highest at 50 °C. On the other hand, setting higher temperatures did not produce a significant change in the yield. Temperature is the most widely varied parameter in MAE investigation. The increase in temperature causes decrease in viscosity and surface tension of the solvent, increasing its matrix penetration power and resulting in enhanced extraction. Based on the observation of this experiment a temperature range of 40–60 °C was selected to be applied for BBD method optimization.

#### 3.1.2. Extraction Time Effect

The extraction time impact on the total extraction yield was investigated by varying the extraction time from 20 to 80 min, keeping the extraction temperature (40 °C) and microwave power (300 W) constant. The result obtained in this experiment clearly indicated that the total extraction yield increases with time of extraction from 20 to 40 min but no significant change was observed in extraction yield beyond 40 min of extraction time (Figure 2B). Hence, a range of extraction time from 35 to 55 min was selected to optimize the extraction method using BBD.

#### 3.1.3. Microwave Power Effect

The microwave power effect on the total extraction yield was investigated by adjusting the microwave power from 50 to 600 W, at constant extraction temperature (40 °C) and extraction time (40 min). It is evident from Figure 2C that the total extraction yield was lowest at microwave power 50 W and maximum at 200 W. In contrast, no significant increase in total extraction yield was observed after increasing the microwave power beyond 200 W. A low or moderate power for a longer time is typically preferred to elude a “bumping” incident generating a high power output. Based on the observation of this experiment, a microwave power ranged from 100 to 300 W was selected to be applied for extraction optimization by BBD method.

### 3.2. HPTLC Analysis of TCME

Various compositions of different solvents were used to develop a suitable mobile phase for parthenolide analysis. After examination of different compositions, *n*-hexane and ethyl acetate in the ratio of 3:1 (*v*/*v*) was chosen as best mobile phase, which afforded a strong parthenolide peak (R_f_ = 0.16; Figure 3A), and it efficiently separated various phytoconstituents present in the TCME (Figure 3B) in the absorbance mode at λ_max_ of 575 nm.

The developed HPTLC method furnished high linearity (*R*^2^ = 0.9928) in 100–700 ng/band of linearity range, and low LOD (28.46 ng) and LOQ (86.24 ng) for parthenolide (Appendix A). The recovery of parthenolide was ranged from 97.64% to 98.97% (Appendix A). The precision for parthenolide at different concentration levels was indicated by % RSD (relative standard deviation) and listed in Appendix A. It was ranged from 1.53% to 1.81% (intra-day) and 1.46% to 1.71% (inter-day). As presented in Appendix A, low values of % RSD (1.79–1.84) indicated that the proposed HPTLC method was robust.

### 3.3. BBD Method Optimization of Extraction Conditions

The ranges for three extraction variables, namely, extraction temperature (*P*_1_), extraction time (*P*_2_) and microwave power (*P*_3_) at three level (+1, 0, −1) for extraction parameters optimization by BBD method, was selected based on the observation of a single-factor experiment.

#### 3.3.1. Statistical Analysis and Model Fitting

In Table 2, the results of 17 experimental combinations of three extraction variables were recorded in terms of percentage parthenolide yield (*R*). These experimental combinations of different extraction variables were carried out to know their impact on parthenolide yield. The results were fitted into Equation (3) (second-order polynomial equation) to generate the following equation with coded factors for *R*:*R_parthenolide_* = 0.8946 + 0.0464 *P*_1_ + 0.0181 *P*_2_ + 0.0067 *P*_3_ − 0.0092 *P*_1_*P*_2_ + 0.0064 *P*_1_*P*_3_ − 0.0087 *P*_2_*P*_3_ − 0.0485 *P*_1_^2^ − 0.018 *P*_2_^2^ − 0.029 *P*_3_^2^.(4)

For this BBD-based experimental design, a quadratic model with *R*^2^ value of 0.9989 was emerged as the best fit model. In Table 3, the regression analysis and response regression equation data for the suggested model have been listed.

The observed *R*^2^ (0.9989) and predicted *R*^2^ (0.9845) values for parthenolide were very close to 1, indicating that observed and predicted values were highly correlated. Also, the adjusted *R*^2^ and predicted *R*^2^ had a difference of less than 2, which was required for the fit model. The determined adequate precision (74.23) was greater than 4, which indicated that the model was fit. In Table 4, the ANOVA results (analysis of variance) for the model terms are listed.

The high F-value (681.78) and low *p*-value (<0.05) for the proposed model suggested that the developed model was significant and there is only a 0.01% chance that an F-value this large could occur due to noise. The F-value of 0.3895 and *p*-value of 0.7677 imply that Lack of Fit is not significant relative to the pure error (>0.05), which suggested a 76.77% chance that a Lack of Fit F-value this large could occur due to noise. Non-significant lack of fit is suitable for the model to be fit and sufficient for predicting the responses.

#### 3.3.2. Effect of Extraction Parameters (*P*_1_, *P*_2_, and *P*_3_) on Parthenolide Yield (*R*)

The contributions of each independent variable [extraction temperature (*P*_1_), extraction time (*P*_2_), and microwave power (*P*_3_)] as well their different interactions on *R* are listed in Table 5. The linear variables (*P*_1_, *P*_2_, *P*_3_), interaction variables (*P*_1_*P*_2_, *P*_1_*P*_3_, and *P*_2_*P*_3_), and quadratic variables (*P*_1_^2^, *P*_2_^2^, and *P*_3_^2^) were found to be significant (*p* < 0.05) and to affect the yield of parthenolide *R*. The *R*^2^ value (0.9989) and % CV (0.2898) indicated good precision and reliability of the experimental values [19]. Figure 4 is the three-dimensional (3D) response surface plot that illustrates the key collaborative effects generated for each pair of factors. Each panel illustrates the impact of two factors on the extraction yields, while the third factor was fixed at the zero level, which was 50 °C for *P*_1_, 45 min for *P*_2_, and 200 W for *P*_3_.

The influences of *P*_1_ and *P*_2_ (Figure 4A), *P*_1_ and *P*_3_ (Figure 4B), and *P*_2_ and *P*_3_ (Figure 4C) on *R* were recorded. As illustrated in Figure 4A, the extraction yield was highest at *P*_1_ of 51.5 °C and *P*_2_ of 50.8 min at a fixed microwave power of 200 W. When *P*_1_ exceeded 51.5 °C, the yield decreased. Figure 4B illustrated the effect of *P*_1_ and *P*_3_ on *R* which demonstrated that the yield was increased significantly at *P*_3_ of 211 W at a fixed extraction time of 45 min. Figure 4C illustrated the impact of *P*_2_ and *P*_3_ interaction on *R* at a fixed extraction temperature of 50 °C, which demonstrated that no significant change was observed in *R* with *P*_3_ increase, but a substantial rise in *R* was observed with an increase in *P*_2_. From this observation, we concluded that the maximal *R* could be extracted from *T. camphoratus* stems using microwave extraction at an extraction temperature of 51.5 °C, an extraction time of 50.8 min, and a microwave power of 211 W.

#### 3.3.3. BBD Method Validation

For the *P*_1_, *P*_2_, and *P*_3_ checkpoints, the yield evaluation was found to be within the detection limits. The experimental value and predicted value of the response were compared to validate *P*_1_, *P*_2_, and *P*_3_ results. The small percentage prediction error facilitated establishing the rationality of the generated polynomial equation and relating BBD model application. The linear correlation plot between actual and predicted values demonstrated a high *R*^2^ value (0.9989), indicating excellent goodness of fit (*p* < 0.0001) (Figure 5).

#### 3.3.4. Optimization and Verification of Microwave-Assisted Extraction Conditions

The selected factors exhibited different effects on parthenolide yield (Table 6). The predicted optimal conditions for parthenolide extraction were as follows: extraction temperature of 51.5 °C (*P*_1_), extraction time of 50.8 min (*P*_2_), and microwave power of 211 W (*P*_3_). The low residual percentage (1.26%) calculated between predicted (0.9157% *w*/*w*) and observed responses (0.9273% *w*/*w*) indicated that the model was reliable.

### 3.4. Cytotoxic Assay of BBD-Run TCME Samples

The in vitro anti-cell proliferative activities of 17 BBD-run *TCME* samples (Runs 1–17) tested against cultured HepG2 and MCF-7 cells exhibited promising cytotoxicity, with estimated IC_50_ values ranging from 33.92 to 71.83 μg/mL and from 39.57 to 75.44 μg/mL, respectively (Figure 6).

Bioactive parthenolide derived from different plant sources are demonstrated to have in vitro anticancer potential on a range of human cancer cell lines. In line with this, the *T. camphoratus* extract obtained by using the BBD optimized microwave extraction conditions exhibited marked cytotoxic effect on HepG2 cells (IC_50_: 30.87 µg/mL) and MCF-7 cells (IC_50_: 35.41 µg/mL) which further supported our HPTLC finding of high parthenolide content (0.9273% *w*/*w*) in TCME.

## 4. Discussion

The effect of different extraction parameters (extraction temperature, extraction time and microwave power) on the parthenolide extraction from the stems of *T. camphoratus* using microwave extraction technique was evaluated and optimized by Box–Behnken design (BBD) of Response Surface Methodology (RSM). Initially, the range for BBD optimization of different extraction parameters was set based on observing single-factors’ effect on the total extraction yield of *T. camphoratus*. The result of single-factor effect revealed that the total yield significantly increased when extraction temperature, extraction time and microwave power reached up to 50 °C, 50 min and 200 W, respectively, and afterward there was no significant increase in the yield was observed upon increase in temperature, time and microwave power. The increased temperature of extraction increases the mass transfer rate and causes higher molecular diffusion, resulting in high extraction of plant material [20]. Similarly increase in microwave power increases the total yield which might be due to an increase in mass transfer driving force. Based on the observation of the single-factor effect on the total yield, the range of extraction temperature (40 °C–60 °C), extraction time (35–45 min), and microwave power (100–300 W) was selected for optimization by BBD method.

During the optimization of different extraction parameters by BBD method, it furnished 17 runs analyzed for the parthenolide content by validated HPTLC method. For BBD analysis, a quadratic model was found as the best fit model. The observed *R*^2^ and predicted *R*^2^ were found close to 1 and their difference was less than 2, indicating a good correlation between them and the model fit. The higher value of adequate precision (>4) suggested that the signal was fair, which could be used to navigate the design space, while low lack of fit value was found not significant, indicating the validity of BBD results. The high model F-value for parthenolide implied that the model was significant. In this study, the low value of percentage residual and considerable value of *R*^2^ supported the high predictive ability of BBD analysis.

The significance of each extraction variable (*P*_1_, *P*_2_, and *P*_3_) on parthenolide extraction and the BBD analysis was evaluated. When *P*_1_ and *P*_2_, *P*_2_ and *P*_3_ interacted, they negatively affected the parthenolide yield. Similarly, the square root of all extraction variables produces a negative impact on parthenolide yield. Only the interactions between *P*_1_ and *P*_3_ produced a positive impact on parthenolide extraction. From the 3D plot, it was clear that *P*_1_ (extraction temperature) had a more significant effect on the parthenolide extraction than the other extraction parameter. The parthenolide content was found maximum at an optimized temperature of 51.5 °C, demonstrating that a lower temperature helped enhance the compound yield. High extraction temperature would result in decreased parthenolide yield. Hence, to extract maximum parthenolide from stems of *T. camphoratus*, the optimum extraction condition was found as 51.5 °C, 50.8 min, and 211 W, respectively.

## 5. Conclusions

The experimental findings suggested that the BBD method is very promising in optimizing the various extraction conditions used in microwave extraction to get the maximum yield of parthenolide from *T. camphoratus* stems. The model prediction can be used to optimize the yield within the limits of the experimental variables. The experimental yield obtained after extracting the sample under optimized extraction conditions [temperature (51.5°C), time of extraction (50.8 min), and microwave power (211 W)] was found as 0.9273% ± 0.0487% *w*/*w*, which agreed closely with the predicted value of 0.9157% *w*/*w*. The quadratic polynomial model was most appropriate concerning parthenolide, and the high values of adjusted *R*^2^ and predicted *R*^2^ indicated good correlation and model fitting, while a high value of signal-to-noise ratio indicated an adequate signal that might be applied to navigate the design space. In the future, this optimized MAE method can be further used to extract parthenolide efficiently from the marketed herbal supplements containing different species of *Tarconanthus*.

## Figures and Tables

**Figure 1 molecules-26-01876-f001:**
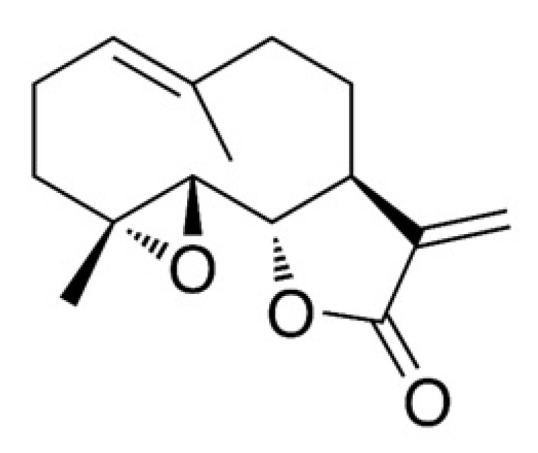
Chemical structure of parthenolide.

**Figure 2 molecules-26-01876-f002:**
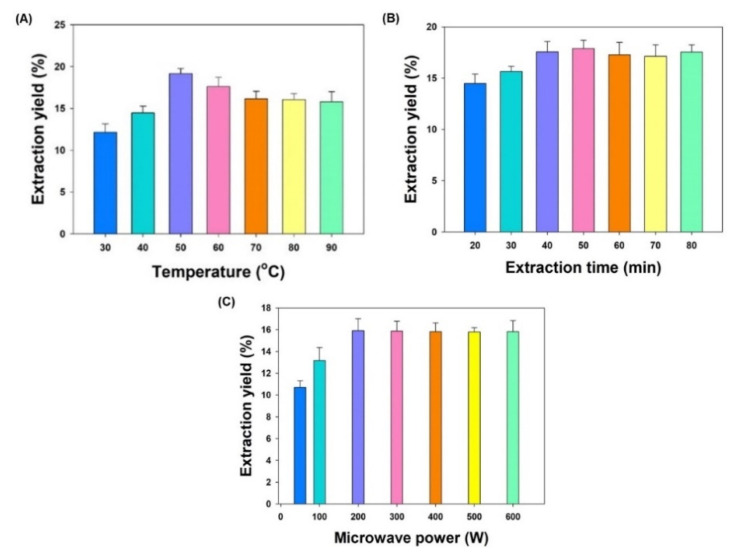
The effects of single factors on the total extraction yield of *TCME*. (**A**) Extraction temperature effect; (**B**) Extraction time effect; (**C**) Microwave power effect. Each value represents a mean ± SD (*n* = 5).

**Figure 3 molecules-26-01876-f003:**
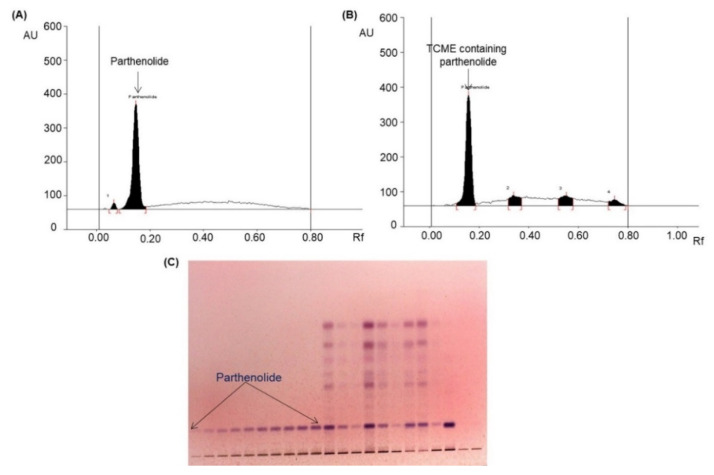
Quantification of parthenolide in BBD-run *TCME sample* by HPTLC [mobile phase—*n*-hexane:ethyl acetate (3:1, *v*/*v*); λ_max_ = 575 nm]. (**A**) Chromatogram of standard parthenolide (R_f_ = 0.16); (**B**) chromatogram of the *TCME* sample (parthenolide, spot 1, R_f_ = 0.16); (**C**) pictogram of TLC plate derivatized with *p*-anisaldehyde in daylight.

**Figure 4 molecules-26-01876-f004:**
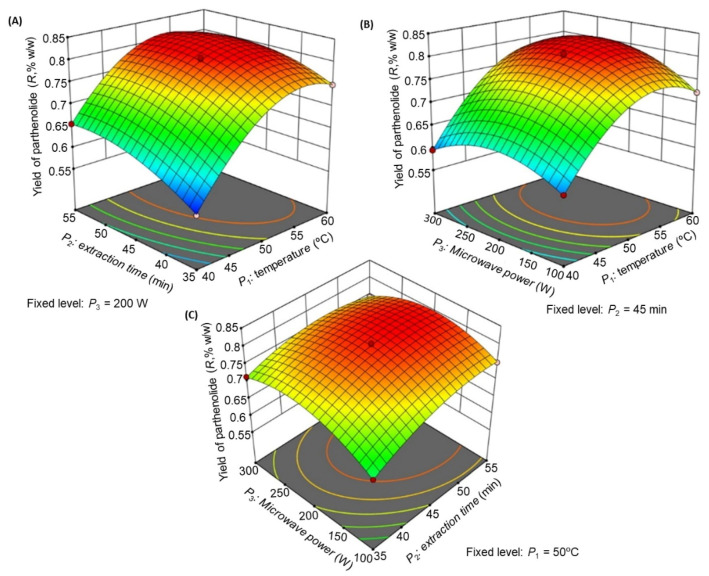
Response surface model 3D plots showing the effects of *P*_1_, *P*_2_, and *P*_3_ on *R*. (**A**) Effect of *P*_1_ and *P*_2_ on *R*, (**B**) effect of *P*_1_ and *P*_3_ on *R*, and (**C**) effect of *P*_2_ and *P*_3_ on *R*.

**Figure 5 molecules-26-01876-f005:**
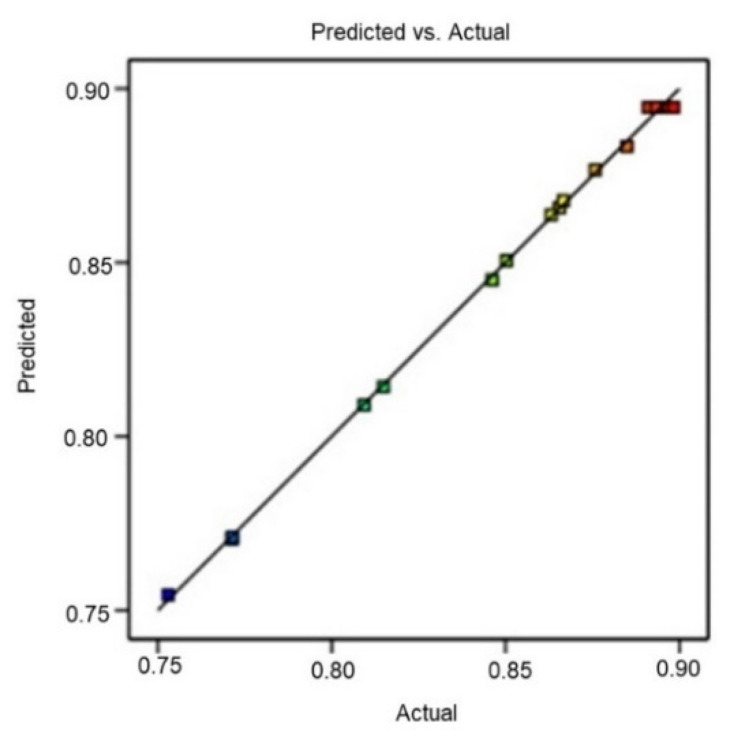
Linear correlation plot between actual and predicted values for *R*.

**Figure 6 molecules-26-01876-f006:**
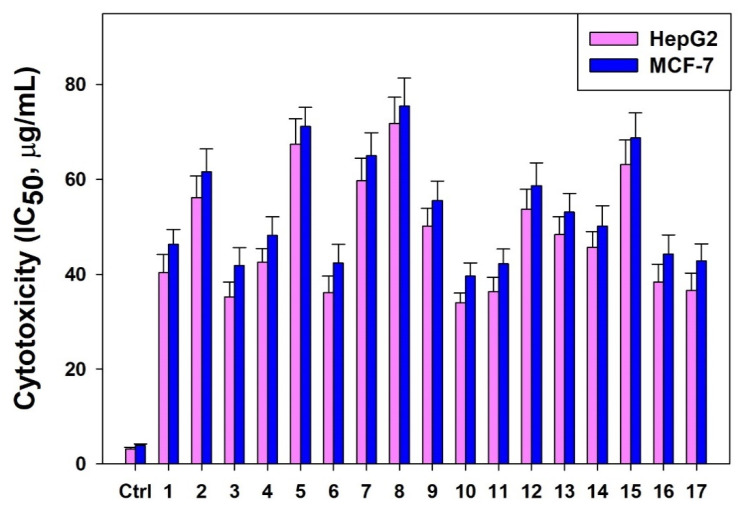
The estimated IC_50_ values (µg/mL) of BBD-run *TCME* samples on HepG2 and MCF-7 cell lines. (ctrl: control).

**Table 1 molecules-26-01876-t001:** Extraction variables selected for Box-Bohnken Design (BBD) optimization.

Independent Variable	Factor Level			Dependent Variable	Goal
	−1	0	+1		
Extraction temperature (°C) (*P*_1_)	40	50	60	Parthenolide yield (% *w*/*w*) (R)	Maximized
Extraction time (min) (*P*_2_)	35	45	55		
Microwave power (W) (*P*_3_)	100	200	300		

**Table 2 molecules-26-01876-t002:** Experimental parameters of Box–Behnken design and result of *R* (parthenolide).

Run	Factor (Coded)			Actual Variables			Parthenolide Yield (*R*)		
	(*P*_1_) (°C)	(*P*_2_) (min)	(*P*_3_) (mL/g)	(*P*_1_) (°C)	(*P*_2_) (min)	(*P*_3_) (W)	Experimental (% *w*/*w*)	Predicted (% *w*/*w*)	Residual
1	1	0	1	60	45	300	0.8758 ± 0.039	0.8767	−0.0009
2	0	−1	−1	50	35	100	0.8149 ± 0.033	0.8143	0.0006
3	0	0	0	50	45	200	0.8967 ± 0.041	0.8946	0.0020
4	0	1	−1	50	55	100	0.8666 ± 0.044	0.8678	−0.0012
5	−1	0	−1	40	35	100	0.7714 ± 0.04	0.7705	0.0009
6	0	0	0	50	45	200	0.8939 ± 0.037	0.8946	−0.0008
7	−1	1	0	40	55	200	0.8093 ± 0.031	0.8090	0.0003
8	−1	−1	0	40	35	200	0.7530 ± 0.048	0.7545	−0.0015
9	1	0	−1	60	45	100	0.8503 ± 0.053	0.8506	−0.0003
10	0	0	0	50	45	200	0.8983 ± 0.058	0.8946	0.0037
11	0	0	0	50	45	200	0.8933 ± 0.051	0.8946	−0.0013
12	0	−1	1	50	35	300	0.8462 ± 0.042	0.8450	0.0012
13	0	1	1	50	55	300	0.8631 ± 0.039	0.8637	−0.0006
14	1	−1	0	60	35	200	0.8654 ± 0.062	0.8657	−0.0003
15	−1	0	1	40	45	100	0.7714 ± 0.022	0.7711	0.0003
16	1	1	0	60	55	200	0.8849 ± 0.054	0.8834	0.0015
17	0	0	0	50	45	200	0.8911 ± 0.038	0.8946	−0.0036

*P*_1_: extraction temperature, *P*_2_: extraction time, *P*_3_: microwave power.

**Table 3 molecules-26-01876-t003:** Regression analysis and response regression equation results for the final proposed model.

Dependent Variables	Source	*R* ^2^	Adjusted *R*^2^	Predicted *R*^2^	SD
*R*	Linear	0.5424	0.4368	0.3041	0.0362
	2FI	0.5640	0.3024	−0.1510	0.0403
	Quadratic	0.9989	0.9974	0.9945	0.0025
	Cubic	0.8791	0.7665	−	−

**Table 4 molecules-26-01876-t004:** ANOVA of the reduced quadratic model for extraction yields of parthenolide.

Dependent Variable	Source	Sum of Squares	Degree of Freedom	Mean Square	F-Value	*p*-Value	Remarks
*R*	Model	0.0372	9	0.0041	681.78	<0.0001	Significant
	Residual	0.0001	7	6.064 × 10^−6^	−	−	
	Lack of fit	9.597 × 10^−6^	3	3.199 × 10^−6^	0.3895	0.7677	Insignificant
	Pure error	0.0001	4	8.212 × 10^−6^	−	−	

**Table 5 molecules-26-01876-t005:** The significance of each response variable effect shown by using the F ratio and *p*-value in the nonlinear second-order model.

Dependent Variable	Independent Variables	SS ^a^	DF ^b^	MS ^c^	F-Value	*p*-Value ^d^
	Linear effects					
	*P* _1_	0.0172	1	0.0172	2842.99	<0.0001
	*P* _2_	0.0026	1	0.0026	430.20	<0.0001
	*P* _3_	0.0004	1	0.0004	58.63	0.0001
	Quadratic effects					
*R*	*P* _1_ ^2^	0.0099	1	0.0099	1632.69	<0.0001
	*P* _2_ ^2^	0.0014	1	0.0014	224.96	<0.0001
	*P* _3_ ^2^	0.0035	1	0.0035	582.25	<0.0001
	Interaction effects					
	*P* _1_ *P* _2_	0.0003	1	0.0003	56.14	0.0001
	*P* _1_ *P* _3_	0.0002	1	0.0002	26.79	0.0013
	*P* _2_ *P* _3_	0.0003	1	0.0003	49.86	0.0002

^a^ Sum of squares; ^b^ Degree of freedom; ^c^ Mean Square; ^d^
*p*-values < 0.05 were considered to be significant.

**Table 6 molecules-26-01876-t006:** Observed and predicted levels for optimal extraction conditions.

Factor		Optimal Level	
*P*_1_ (°C)		51.5	
*P*_2_ (min)		50.8	
*P*_3_ (W)		211 W	
Response	Predicted (% *w*/*w*)	Experimental (% *w*/*w*, *n* = 3)	Residual (%)
Parthenolide (% *w*/*w*)	0.9157	0.9273 ± 0.0487	1.26

Residual (%) = (observed value − expected value)/expected value × 100.

## Data Availability

All the data generated and used is presented in this article.

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
