# Peer review of "Box–Behnken Design (BBD)-Based Optimization of Microwave-Assisted Extraction of Parthenolide from the Stems of Tarconanthus camphoratus and Cytotoxic Analysis"

_molecules, 2021, doi:10.3390/molecules26071876_

Round 1

Reviewer 1 Report

In this work, Authors have developed a BBD-based optimization of MAE of parthenolide from Traconanthus camohoratus and they have also carried out the cytotoxic analysis of the extract. The work presents great novelty and scientific interest. However, there are some minor aspects that should be revised before the whole work reaches the standards to be published in Molecules.

In general terms the use of the language and grammar should be revised. The use of the third and first person should be checked and impersonal expression should be included. As examples:  Line 15 “prompet us”, Line 83 “the authors had”…., Line 180 “experiments was…”

Section 2.1:

  • Had been done any sample preparation here? Collection, drying, sampling, storage?

Section “Apparatus and reagents”:

  • This should be Section 2.2. Revise this aspect along the manuscript (2.3 is also repeated).
  • “materials “should be also included in the title taking into account the information included in this section.
  • Analytical standards and their purity should be included.
  • There are some solvent used in the procedure such as DMSO that are

Section 2.2:

  • Line 110 the uncertainty of the measurement of the sample should be included.

Section 2.3:

  • If the extract is dried, how is possible to applied the samples in the HPTLC system? Have the authors done any solvent exchange?

Section 2.3.3:

  • LOD and LOQ should be defined.

Section BBD Experimental design:

  • Why author carried out the step by step approximation. Why not directly used the BBD. This is an important aspect that should be clearly explained and described taking into account the aim of the manuscript.
  • Fixed conditions, used response, replicated and concentration is data that should be included in the text. Not only in figures.
  • Standards deviation of the replicates should be included in the figures.

Section 3.2:

  • Is there any result that support the remark “various compositions of different solvents used to develop a suitable mobile phase… the best mobile phase …”

Reviewer 2 Report

In manuscript molecules-1142074, the authors have presented a method for the maximization of the yield of parthenolide from Tarconanthus camphoratus using microwave-assisted extraction.  This manuscript presents not only a refinement for microwave extraction of parthenolide from T. camphoratus, but also presents a procedure for maximizing the performance for extraction of other desirable phytochemicals from other plant species.

Lines 383-386:  “…we demonstrated the marked cytotoxic effect of a T. camphoratus extract obtained by using the optimal extraction conditions on HepG2 cells (IC50: 30.87 µg/ml) and MCF-7 cells (IC50: 35.41 µg/ml).  The in vitro results further endorse the HPTLC data on high contents of parthenolide in TCME obtained through the optimized extraction conditions.”  This statement is misleading.  There seems to be very little correlation between cytotoxicity and parthenolide yield.  For example, sample 10, with a parthenolide yield of 0.898%, showed lower cytotoxic activities against both cell lines than sample 5 (0.771% parthenolide yield).  Some additional comment is needed here.

Minor editorial corrections:

Line 37:  Chrysanthemum parthenium (feverfew) [add the “feverfew” since it is used later on].

Lines 47 and 81:  Do not capitalized “feverfew”

Lines 79-80:  Correct “Several research has been reported…”.  Suggestion:  “Several research groups have reported...”

Line 132:  Do not capitalized “parthenolide”
